# Effect of the Resveratrol Rice DJ526 on Longevity

**DOI:** 10.3390/nu11081804

**Published:** 2019-08-05

**Authors:** Md. Saidul Islam, Yan Yan Jin, Hea-Jong Chung, Hyeon-Jin Kim, So-Hyeon Baek, Seong-Tshool Hong

**Affiliations:** 1Department of Biomedical Sciences and Institute for Medical Science, Chonbuk National University Medical School, Jeonju, Chonbuk 54907, Korea; 2BDRD Institute, JINIS Biopharmaceuticals Co., Bongdong, Wanju, Jeonbuk 55321, Korea; 3Department of Well-Being Resources, Sunchon National University, Suncheon, Jeonnam 57922, Korea

**Keywords:** resveratrol, lifespan, the resveratrol rice DJ526, *Drosophila*

## Abstract

Resveratrol is the best-known chemical for extending the lifespan of various organisms. Extensive recent research has shown that resveratrol can extend the lifespan of single-celled organisms, but its effects on the extension of animal lifespans are marginal. Despite the limited efficacy of pure resveratrol, resveratrol with the endogenous property of the DJ rice in the resveratrol rice DJ526 previously showed profound health benefits. Here, we report that the resveratrol rice DJ526 markedly extended the lifespan of the fruit fly *Drosophila melanogaster* by as much as 41.4% compared to that of the control. The resveratrol rice DJ526 also improved age-related symptoms such as locomotive deterioration, body weight gain, eye degeneration and neurodegeneration in *D. melanogaster* upon aging. This result shows the most significantly improved lifespan in animal experiments to date, meaning that the resveratrol rice DJ526 will assist in the development of a therapeutic agent for longevity or addressing age-related degeneration.

## 1. Introduction

Human aging occurs through the gradual deterioration of the body over time. Naturally, this deterioration of the body makes aging the greatest risk factor for most human diseases [1,2]. Although addressing aging has been the primary scientific pursuit of modern health care, there is not much progress in understanding the aging process and anti-aging management. In this context, it is not surprising that many serious modern researchers have aimed to find a solution for anti-aging management.

Resveratrol (3, 5, 4′-trihydroxy-trans-stilbene) is a polyphonic bioflavonoid phytonutrient and an antioxidant produced by plant roots and fruits [3,4]. Although the identity of resveratrol has been well known for decades, the compound did not receive scientific attention until it was shown to extend the lifespan of *Saccharomyces cerevisiae* by stimulating the silent information regulator (Sir) [5,6,7,8]. Despite its excellent anti-aging effect in yeast, the therapeutic efficacy of resveratrol showed diminished reliability in animal studies [9,10,11]. Resveratrol also extended the lifespan of *Caenorhabditis elegans* through a Sir-2.1-dependent mechanism that inhibited the action of endoplasmic reticulum (ER) stress genes [12]. In contrast, resveratrol did not extend the lifespan of *C. elegans* under normal conditions but extended the lifespan under acute oxidative stress conditions in a dose-dependent manner [13]. Besides, resveratrol and other sirtuin-activating compounds (STACs) extended the lifespan of *C. elegans* and *Drosophila melanogaster* in a Sir2-dependent manner by mimicking caloric restriction [14], while another study reported that dietary resveratrol supplementation did not exert its beneficial lifespan extension effect on *D. melanogaster* [15,16]. In an animal model, resveratrol altered the physiology of mice on a high-calorie diet, showing improved health and a slight extension of lifespan [11,17,18]. By contrast, long-term resveratrol administration failed to extend the lifespan of mice while it slowed down age-related degeneration with similar changes to the gene expression patterns induced by dietary restriction [19,20]. The aforementioned findings have been suggested to indicate that resveratrol extends the lifespan of single-celled organisms such as *Saccharomyces cerevisiae*, but its effects on lifespan extension in animals were very limited.

Previously, we developed the resveratrol rice DJ526 by transferring the resveratrol biosynthesis gene, *stilbene synthase*, from the *Arachis hypogaea* (peanut) variety Palkwang into *Oryza sativa* (rice) japonica variety Dongjin (DJ), which accumulated a significant amount of resveratrol, at 1.4–1.9 μg/g, in its grains [21,22,23,24]. The resveratrol rice DJ526 showed significant health-benefits on obesity and related metabolic syndrome through the synergistic effect of resveratrol with the endogenous property of the DJ rice compared to the resveratrol or control, which had minimal effects [21,22,23,24]. The therapeutic efficacies of the resveratrol rice DJ526 on obesity and related metabolic syndrome were comparable to those of typical pharmaceutical drugs [21,22,23,24].

Considering that the original efficacy of resveratrol is anti-aging, it would be worthwhile to investigate the anti-aging efficacy of resveratrol rice DJ526 as human food. In our recent study, we investigated the longevity effect of DJ526 callus induced from the mature seeds of the resveratrol DJ526 rice, demonstrating that the callus of the resveratrol rice DJ526 could extend the lifespan of *Drosophila melanogaster* [25]. In continuation of our longevity study with DJ526, it is worthwhile to investigate the edible part of rice, the grains from DJ526 rice. In this study, we investigate the anti-aging effects of the resveratrol rice DJ526 using grains to feed *D. melanogaster*. Our experimental results show that the resveratrol rice DJ526 markedly extends *D. melanogaster* lifespan by as much as 41.4% compared to the control. This degree of lifespan extension of an experimental animal is the most significant extent to date compared to other therapeutic agents. We believe that the resveratrol rice DJ526 deserves further attention and study for its possible therapeutic use for longevity or age-related diseases, as well as its contribution to elucidating the aging process.

## 2. Materials and Methods

### 2.1. Drosophila Strains and Maintenance

The wild-type Harwich strain (FBst0004264) of *D. melanogaster* was obtained from the Bloomington *Drosophila* Stock Center (Indiana University, Bloomington, IN, USA) and the wild-type ORR strain of *D. melanogaster* was provided by Isaac A. Adedara (Federal University of Santa Maria, Santa Maria, RS, Brazil). The *D. melanogaster* individuals used in these experiments were routinely maintained at 18 °C on standard cornmeal media in a 60% humidified incubator with a 12 h light–12 h dark cycle.

### 2.2. Drosophila Experiments

The *D. melanogaster* maintained on the standard cornmeal media at 25 °C were divided into five groups to transfer onto standard cornmeal media (Ctrl), cornmeal media supplemented with resveratrol (RS) media, cornmeal media supplemented with DJ rice (DJ), DJ media supplemented with resveratrol (DJRS), and cornmeal media supplemented with resveratrol rice DJ526 (DJ526) for laying eggs (Appendix A), and the fly larvae were maintained in the media at 25 °C. Within a few hours of eclosion, the flies were collected under brief CO_2_ anesthesia and incubated at 25 °C for 48 h for maturation. The mature flies were transferred to the respective diets as indicated above and cultured at 18 °C in a humidified incubator under a 12 h light–12 h dark cycle. After the fly maturation, the rest of the fly experiments were performed at 18 °C. Each of the 150 adult male and 150 adult female flies were used for the lifespan assay. A vial large enough for 50 flies was used to avoid over-crowding. The flies were transferred to fresh vials with fresh media every 4 days until all flies were dead. The number of dead flies was recorded, and the survival data were analyzed using the Kaplan–Meier method. The Kaplan–Meier estimate is the non-parametric maximum likelihood of survival at a given time point *S(t)* as follows:(1)$$S\left(t\right)=\underset{\mathrm{ti}<t}{\prod }\frac{\mathrm{ni}-\mathrm{di}}{\mathrm{ni}}.$$
where *ni* is the number of survivors minus the number of censored cases and *di* is the number of deaths at time point *ti* [26].

### 2.3. Locomotion Assay

The adult male and female flies post-eclosion with the same starting age were on the experimental diets as indicated. For the locomotion assays, 15 flies from each group were maintained in a plastic *Drosophila* culture vial and the media were changed every 4 days. The flies in the plastic tube were anaesthetized with CO_2_ by being placed into 15-mL empty plastic tubes, which were bunged with cotton wool to prevent escape, at room temperature. The flies were recovered from CO_2_ exposure for 30 min before the assay. After gentle tapping, the flies at the bottom of the tube were allowed to climb a height of 10 mL within 30 s for the observation of their climbing performances. The number of flies at the top, above the 10-mL mark of the tube, and at the bottom, below the 2-mL mark of the tube, was recorded after 30 s. Three trials were performed for each respective diet at each time point. The climbing abilities of the flies were observed at the 10th, 30th, 60th and 90th days post-eclosion. The total number of flies used in each group was 100. The performance index (PI) was calculated for each wild-type *Drosophila* group of 15 flies using the formula PI = 0.5 × (*n*_total_ + *n*_top_ − *n*_bottom_)/*n*_total_, where *n*_total_ is the total number of flies, *n*_top_ is the total number of flies at the top, and *n*_bottom_ is the total number of flies at the bottom [27].

### 2.4. Body Weight Measurements

Newly eclosed fresh adult male and female flies were fed the experimental media as indicated. The body weights of the individual adult flies were measured at the 10th, 30th, 60th and 90th days post-eclosion.

### 2.5. Light Microscopy of the Drosophila Eye

Adult male and female flies were fed respective diets and maintained as described above. The flies were collected on the indicated days post-eclosion. Ten male and 10 female flies of the WT ORR and WT Harwich strains from each respective media were anaesthetized with CO_2_ and transferred into the Eppendorf tube. The flies were then frozen at −80 °C for 1 h and mounted on their sides, and their eyes were examined using a dissecting light microscope (Amscope, ZM-4TW3-FOR-8M, Irvine, CA, USA) equipped with an Olympus SZ51 lamp; the eye images were captured with a microscopic camera (Amscope, MU-1000). The eye images of flies were analyzed for each respective diet at the indicated days post-eclosion.

### 2.6. Histological Examination of the Drosophila Brains

The adult male and female flies were fed respective types of diets and maintained as described above. The flies were collected on the 10th, 30th, 60th and 90th days post-eclosion for histological examination. After anesthesia with CO_2_, the flies were kept at −80 °C for 1 h. The fly heads were collected and fixed in 10% neutral buffered formalin (Sigma Aldrich, St. Louis, MO, USA) at room temperature, followed by embedding in paraffin using standard histological procedures. The embedded heads were sectioned at 6 µm for histological examination. The paraffin in the brain section on glass microscopic slides was removed through washing with hot water. Then, the slides were air-dried and baked overnight at 65 °C. The brain sections were stained with haematoxylin and eosin. The stained brain images were taken under a slide scanner microscope by Apero Scan Scope FL (Leica Biosystems, Nussloch, Germany) at a magnification of 10×; scale bars, 200 μm.

### 2.7. Statistical Analysis

An analysis of the survival data was performed using the Kaplan–Meier method with data preparation using Graph Pad Prism version 8.1.2 software (GraphPad Software, Inc. San Diego, CA, USA). All comparisons were made with log-rank tests (Figure 1). The statistical analysis was expressed as the means ± standard deviation (s.d.) as indicated. The significant differences between the two groups were analyzed with an unpaired Student’s *t*-test, being considered statistically significant if *p* < 0.05. The statistical significance was shown as * *p* < 0.05, ** *p* < 0.01 and *** *p* < 0.001 from three independent experiments.

## 3. Results

### 3.1. The Resveratrol Rice DJ526 Dramatically Extended the Median Lifespan of D. melanogaster

To evaluate the efficacy of the resveratrol rice DJ526 on lifespan extension in fly, two different wild-type (WT) strains of *D. melanogaster*, ORR and Harwich, were used in this study. Feeding experiments were conducted by feeding WT *Drosophila* five different diets; a control diet with standard cornmeal diet of *Drosophila*; a DJ diet, in which 50% of the cornmeal in the standard cornmeal diet was replaced with the DJ rice; a DJ526 diet, in which 50% of the cornmeal was replaced with the resveratrol rice DJ526; an RS diet, in which 31.54 µg/L resveratrol, the equivalent amount found in the resveratrol rice DJ526, was supplemented to the standard cornmeal diet; a DJRS diet, in which 50% of the cornmeal was replaced with the DJ rice with supplementation of resveratrol at 31.54 µg/L (Appendix A).

The effect of the resveratrol rice DJ526 on lifespan was evaluated by recording the dead flies with aging (Figure 1). In concordance with previous publications on the effect of resveratrol, the median lifespan increment of the RS groups (*D. melanogaster* was given the diet containing an equivalent amount of resveratrol as that of the DJ526 group) was not significant compared to the control groups (*D. melanogaster* receiving a standard cornmeal diet). The survival rate differences between each group became evident from the 30th day of the feeding experiments. The median lifespans of the ORR flies in the DJ526 groups were 152 days for males and 156 days for females, which were most extended compared to the control groups (108 days for males and 112 days for females), RS groups (116 days for both males and females), DJ groups (140 days for males and 144 days for females), and DJRS groups (144 days for males and 148 days for females). In log-rank tests, the resveratrol rice DJ526 significantly extended the median lifespan of ORR flies compared to the control groups (*p* < 0.0001 for both male and female), RS groups (*p* < 0.0001 for both male and female) DJ groups (*p* = 0.0228 for male and *p* = 0.0259 for female), and DJRS groups (*p* = 0.1074 for male and *p* = 0.4557 for female) (Figure 1A,B).

Similar to the ORR flies, the median lifespans of the Harwich flies in the DJ526 groups were 164 days for both males and females, which was extended most compared to the control groups (116 days for both males and females), RS groups (120 days for males and 128 days for females), DJ groups (156 days for both males and females), and DJRS groups (156 days for males and 160 days for females). In addition, log-rank tests showed that the resveratrol rice DJ526 significantly extended the median lifespan of Harwich flies also, compared to the control groups (*p* < 0.0001 for both male and female), RS groups (*p* < 0.0001), DJ groups (*p* = 0.0721 for male and *p* = 0.0219 for female only), and DJRS groups (*p* = 0.0788 for male and *p* = 0.0794 for female) (Figure 1C,D). Overall, the resveratrol rice DJ526 increased the median lifespan by as much as 40.7% for ORR *D. melanogaster* and 41.4% for Harwich *D. melanogaster,* compared to the control (Figure 1). Overall, the maximum lifespan of the DJ526 group flies was increased significantly (Figure 1). These results clearly showed that the interaction of resveratrol and the endogenous characteristics of DJ rice in the resveratrol rice DJ526 synergistically extended the lifespan of *D. melanogaster*.

### 3.2. The Resveratrol Rice DJ526 Ameliorated the Locomotive Deterioration of D. melanogaster during Age Progression

Since a paucity of locomotion is a typical symptom of aging, we tested the flies’ locomotive ability by measuring *D. melanogaster* climbing. The climbing ability of *D. melanogaster* was measured over their age progression to assess the effect of the resveratrol rice DJ526 (Figure 2). Its effect was noted starting from the 30th day in both *D. melanogaster* strains. The DJ526 groups of ORR strain climbed the test tube 1.52 times better than the control group, 1.42 times better than the RS group, 1.16 times better than the DJ group and 1.08 times better than the DJRS group at the 90th day after feeding. Also, the DJ526 groups of Harwich strain climbed the test tube 1.64 times better than the control group, 1.52 times better than the RS group, 1.21 times better than the DJ group and 1.11 times better than the DJRS group at the 90th day after feeding. These results indicated that the resveratrol rice DJ526 helped to ameliorate locomotive deterioration in *D. melanogaster* upon aging to maintain their locomotive ability, supporting the lifespan increment data in Figure 1.

### 3.3. The Resveratrol Rice DJ526 Caused D. melanogaster to Maintain a Healthy Body Weight during Age Progression

Considering that weight gain is an important indicator of animal aging, we investigated the effect of the resveratrol rice DJ526 on the body weight of *D. melanogaster*. The body weights of all the fly groups gradually increased with age progression (Figure 3). However, as expected from its anti-aging property, the flies in the DJ526 groups did not show excessive weight gain during age progression (Figure 3). The body weights of *D. melanogaster* ORR in the DJ526 groups were 1.64 ± 0.03 mg for male flies and 1.74 ± 0.01 mg for female flies at the 90th day after feeding, which were significantly lower than those of the control groups (1.91 ± 0.07 mg for males and 2.14 ± 0.09 mg for females), RS groups (1.74 ± 0.04 mg for males and 2.11 ± 0.09 mg for females), DJ groups (1.70 ± 0.02 mg for males and 1.82 ± 0.03 mg for females), and DJRS groups (1.70 ± 0.01 mg for males and 1.79 ± 0.02 mg for females).

Similarly, the body weights of *D. melanogaster* Harwich in the DJ526 groups were 1.58 ± 0.01 mg for male flies and 1.74 ± 0.01 mg for female flies at the 90th day after feeding, which were significantly lower than the body weights of the control groups (1.74 ± 0.05 mg for males and 2.00 ± 0.06 mg for females), RS groups (1.69 ± 0.03 mg for males and 1.98 ± 0.06 mg for females), DJ groups (1.67 ± 0.08 mg for males and 1.83 ± 0.03 mg for females), and DJRS groups (1.65 ± 0.02 mg for males and 1.82 ± 0.03 mg for females). Overall, the resveratrol rice DJ526 provided prominent health benefits relating to the maintenance of a healthy body weight compared to other groups of flies, supporting our recent study with DJ526 callus [25].

### 3.4. The Resveratrol Rice DJ526 Inhibited Eye Degeneration in D. melanogaster during Age Progression

*Drosophila* eye is known as an ideal model for investigating morphogenic changes, cell fate specification, and patterning [28]. We noticed that the morphological characteristics of the *Drosophila* eye accurately represented the degree of aging. The morphological characteristics of eye damage, roughness and the loss of eye pigment, were used to assess the severity of aging (Figure 4). The morphological observation of the *Drosophila* eyes clearly showed that the resveratrol rice DJ526 prevented eye degeneration during age progression. It should be noted that the eyes of WT flies, both the ORR and Harwich flies, under the DJ526 diet were rarely damaged, even at the 90th day after feeding, which was significantly different from the other groups of flies.

### 3.5. The Resveratrol Rice DJ526 Ameliorated Neurodegeneration in D. melanogaster during Age Progression

Since DJ526 inhibited aging and ameliorated the aging symptoms in *D. melanogaster* (Figure 1, Figure 2, Figure 3 and Figure 4), we also investigated whether the resveratrol rice DJ526 inhibits age-related neurodegeneration with age progression as well. The brains of each group of flies were isolated over the time course of the experiments, and we examined fly brain slices after haematoxylin and eosin (H&E) staining. In the control groups, the vacuolar lesions became widespread as the fly age progressed, which can be used as a critical pathological marker of neurodegeneration in *D. melanogaster* (Appendix A, Figure 5 and Figure 6). While the vacuolar lesions also appeared gradually and spread with age progression in the RS, DJ, and DJRS groups, the brains of the DJ526 group had significantly fewer vacuolar lesions compared to the other groups. The histological differences between each group became dramatically different at the 90th day after feeding. The brains of both fly strains in the DJ526 groups had few vacuolar lesions and maintained generally healthy brain integrity compared to the other groups. Additionally, it is notable that the brains of the RS, DJ, and DJRS groups showed fewer vacuolar lesions and maintained better brain integrity than the control group. The histological observation could be concluded to indicate that the resveratrol rice DJ526 offered excellent protection against neurodegeneration by suppressing the aging process.

## 4. Discussion

The biological process of aging has been well defined. The aging process, however, has been poorly understood with no effective prevention, treatment or therapy for aging [2,29]. Among the various attempts to increase the lifespan, caloric restriction (CR) has been the most effective in expansion of lifespan in diverse species from yeast to mammals [30,31,32,33]. Interestingly, resveratrol administration has been shown to mimic transcriptional aspects of CR in mice [19,34]. That characteristic of resveratrol gives us new hope for developing a longevity drug. However, resveratrol did not increase the lifespan despite delaying aging-related physiological deteriorations by modulating the gene expression of energy metabolism by mimicking CR [3,19,35], which means that resveratrol itself is not sufficient as a therapeutic drug candidate for improving longevity or suppressing aging-related degenerative diseases.

Although the benefits of supplementary resveratrol are limited in animal experiments, our previous studies indicated that the transgenic resveratrol rice DJ526 provided unexpectedly high beneficial health effects through a synergistic mechanism for treating obesity and related metabolic syndrome [21,22,23,24,36]. However, those therapeutic efficacies have not been achieved through pure supplementary resveratrol treatment. This study showed that feeding *D. melanogaster* with the resveratrol rice DJ526 extended the fly lifespan by as much as 41.4%. Although various studies are reporting the extension of the *D. melanogaster* lifespan [37,38,39,40,41,42,43], no chemicals or nutrients have been successfully extended the lifespan of *D. melanogaster* by as much as the resveratrol rice in this work.

The aging process is accompanied by the degeneration of almost all the tissues and organs, as well as the increase in the body weight [1,44,45]. This study showed that the resveratrol rice DJ526 not only increased the lifespan but also ameliorated the degeneration of the eye and brain of *D. melanogaster*. Also, the resveratrol rice DJ526 caused the flies to maintain a healthy body weight and locomotion. Considering that decreased locomotion and increased body weight represent important indicators of aging, it would be reasonable to conclude that the resveratrol rice DJ526 has a clear anti-aging effect. Overall, the resveratrol rice DJ526 could be an ideal functional food for treating and/or preventing aging, as well as various age-related diseases such as obesity, metabolic syndrome, etc.

In contrast to resveratrol rice DJ526 diet, the supplementation of resveratrol had almost no effect on its lifespan extension in *D. melanogaster* (Figure 1) despite its slight improvement in other age-related deterioration (Figure 2, Figure 3, Figure 4, Figure 5 and Figure 6). These experimental results are in good agreement with previous studies showing that resveratrol supplementation has almost no effect or insignificant improvements on the lifespan of *D. melanogaster* [15,16,46]. In this context, this work and our previous research on resveratrol rice DJ526 shines a light on the new therapeutic candidate for anti-aging and longevity. In addition, our experimental results suggest that exploring the synergistic interaction of bioactive chemicals or nutrients in vivo could provide new insights in the development of therapeutic agents.

Despite its seriousness and interest to society, a therapeutic agent that could slow the aging process remains undeveloped. Many studies have reported lifespan extensions in animal experiments in ranges of approximately 10%. Among the reported chemicals or nutrients extending the lifespans of the animals, rapamycin showed the greatest efficacy. However, rapamycin only extended animal lifespans from 9.3 to 16% depending on the experiments [47,48,49]. Although rapamycin impressively increases the lifespan, the efficacy of rapamycin on lifespan extension is still within an insignificant range of approximately 10%, which poses the question as to whether the inhibition of the mTOR pathway by rapamycin [50,51] is a key pathway in anti-aging or simply an anti-aging-related pathway. In this context, this work would shed light on understanding the aging process and developing an anti-aging therapeutic agent.

## 5. Conclusions

The resveratrol rice DJ526 is a genetically modified crop accumulated transgenic resveratrol in its grains, which provides upgrade potential against aging in wild-type ORR and Harwich *D. melanogaster*. Therefore, the resveratrol rice DJ526 leads to the generation of unexpectedly beneficial health effects, which can be used to extend the lifespan of *D. melanogaster* and prevent and/or treat various age-related diseases, bolstering hope that it may one day become possible to extend human lifespan.

## Figures and Tables

**Figure 1 nutrients-11-01804-f001:**
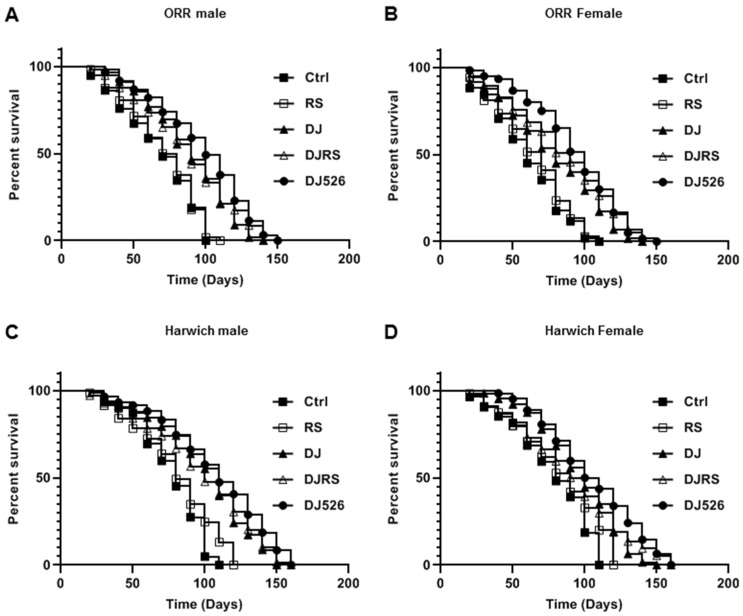
The resveratrol rice DJ526 increased the lifespan of *Drosophila melanogaster*. (**A**) ORR males, (**B**) ORR females, (**C**) Harwich males and (**D**) Harwich females. Ctrl represents standard cornmeal medium; RS represents cornmeal medium supplemented with resveratrol at 31.54 µg/L, the equivalent amount of resveratrol found in the DJ526 media; DJ represents the medium in which 50% of cornmeal was replaced with Dongjin rice; DJRS represents the medium in which 50% of cornmeal was replaced with Dongjin rice and supplemented with resveratrol at 31.54 µg/L; and DJ526 represents the medium in which 50% of the cornmeal was replaced with the resveratrol rice DJ526 (Appendix A). For the lifespan assay, the survival rate of 150 flies from each group was monitored with medium change every 4 days. Comparisons were made using log-rank tests. The *p* values (log-rank tests) for each strain and each sex were as follows. (**A**) ORR male flies: DJ526 versus Ctrl (*p* < 0.0001), RS (*p* < 0.0001), DJ (*p* = 0.0228), and DJRS (*p* = 0.1074), respectively. (**B**) ORR female flies: DJ526 versus Ctrl (*p* < 0.0001), RS (*p* < 0.0001), DJ (*p* = 0.0259), and DJRS (*p* = 0.4557), respectively. (**C**) Harwich male flies: DJ526 versus Ctrl (*p* < 0.0001), RS (*p* < 0.0001), DJ (*p* = 0.0721), and DJRS (*p* = 0.0788), respectively. (**D**) Harwich female flies: DJ526 versus Ctrl (*p* = 0.0001), RS (*p* < 0.0001), DJ (*p* = 0.0219), and DJRS (*p* = 0.0794), respectively. The percentage of surviving flies is shown along with the maximum lifespan in each group (*n* = 150).

**Figure 2 nutrients-11-01804-f002:**
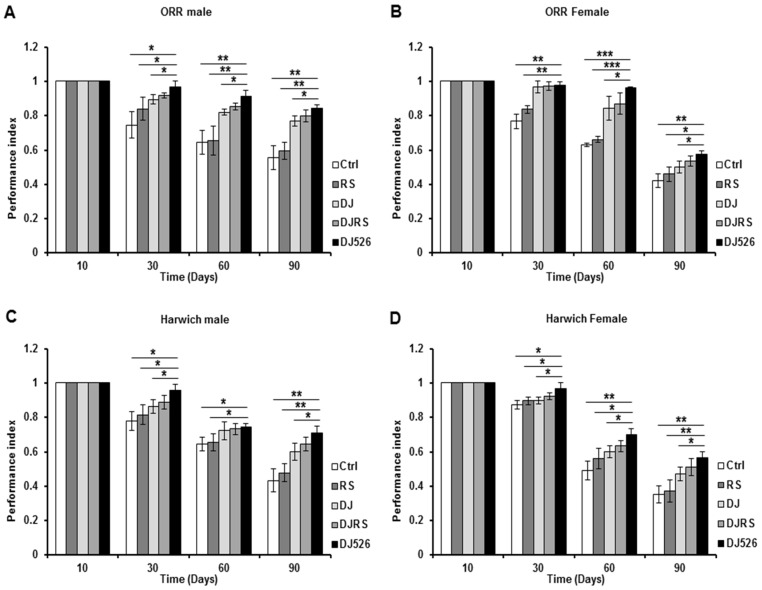
The resveratrol rice DJ526 improved the locomotion activity of *D. melanogaster*. (**A**) ORR males, (**B**) ORR females, (**C)** Harwich males and (**D**) Harwich females with age progression. Ctrl represents standard cornmeal medium; RS represents cornmeal medium supplemented with resveratrol at 31.54 µg/L, the equivalent amount of resveratrol found in the DJ526 media; DJ represents the medium in which 50% of cornmeal was replaced with Dongjin rice; DJRS represents the medium in which 50% of cornmeal was replaced with Dongjin rice and supplemented with resveratrol at 31.54 µg/L, and DJ526 represents the medium in which 50% of the cornmeal was replaced with the resveratrol rice DJ526 (Appendix A). Fly locomotor activity was observed as indicated until 90th days post-eclosion and was indicated as the performance index. Statistical significance was analyzed with an unpaired Student’s *t*-test and indicated as * *p* < 0.05, ** *p* < 0.01, and *** *p* < 0.001 from three independent experiments *(n* = 15).

**Figure 3 nutrients-11-01804-f003:**
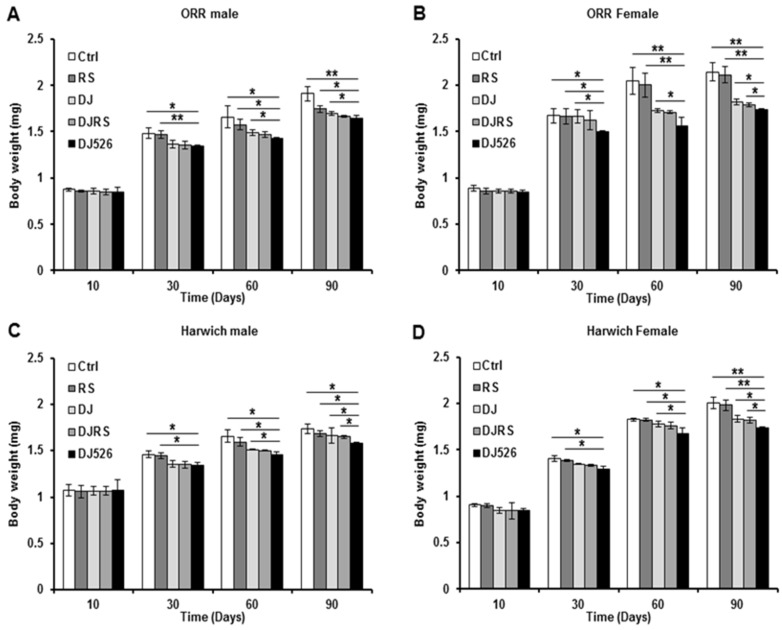
The effects of the resveratrol rice DJ526 on changes in the body weights of *D. melanogaster*. (**A**) ORR males, (**B**) ORR females, (**C**) Harwich males and (**D**) Harwich females with age progression. Ctrl represents standard cornmeal medium; RS represents cornmeal medium supplemented with resveratrol at 31.54 µg/L, the equivalent amount of resveratrol found in the DJ526 media; DJ represents the medium in which 50% of cornmeal was replaced with Dongjin rice; DJRS represents the medium in which 50% of cornmeal was replaced with Dongjin rice and supplemented with resveratrol at 31.54 µg/L, and DJ526 represents the medium in which 50% of the cornmeal was replaced with the resveratrol rice DJ526 (Appendix A). The body weights were measured at the 10th, 30th, 60th and 90th days post-eclosion. Statistical significance was analyzed with an unpaired Student’s *t*-test and indicated as * *p* < 0.05 and ** *p* < 0.01from 3 independent experiments *(n* = 30).

**Figure 4 nutrients-11-01804-f004:**
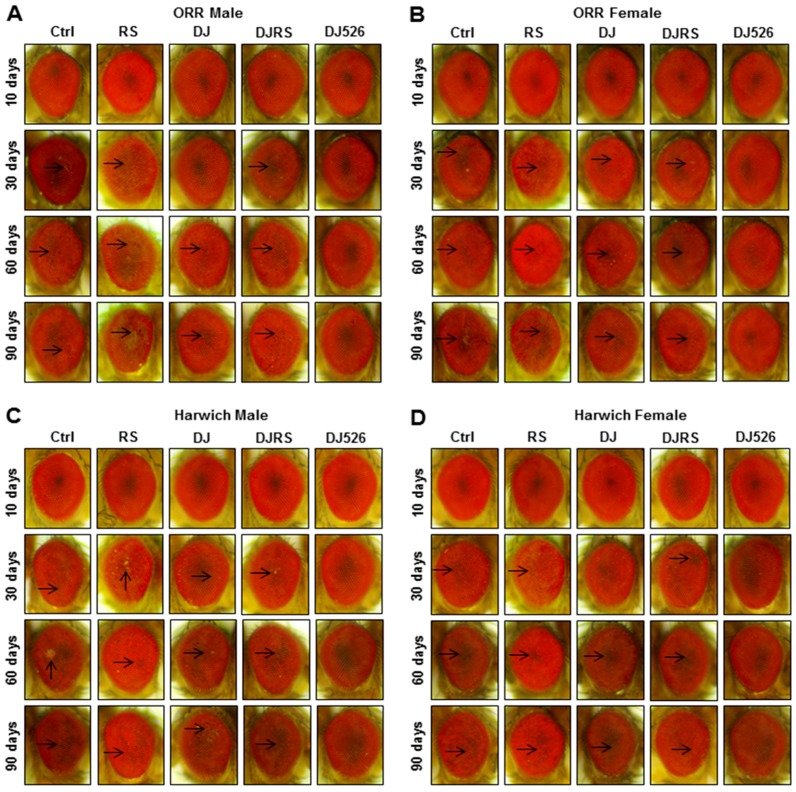
The resveratrol rice DJ526 suppressed the developmental eye defects in *D. melanogaster*. (**A**) ORR males, (**B**) ORR females, (**C**) Harwich males and (**D**) Harwich females with age progression. Ctrl represents standard cornmeal medium; RS represents cornmeal medium supplemented with resveratrol at 31.54 µg/L, the equivalent amount of resveratrol found in the DJ526 media; DJ represents the medium in which 50% of cornmeal was replaced with Dongjin rice; DJRS represents the medium in which 50% of cornmeal was replaced with Dongjin rice and supplemented with resveratrol at 31.54 µg/L, and DJ526 represents the medium in which 50% of the cornmeal was replaced with the resveratrol rice DJ526 (Appendix A). Light microscopy studies of the *Drosophila* compound eyes were performed at the 10th, 30th, 60th and 90th days post-eclosion, and the eye damages are indicated as arrows.

**Figure 5 nutrients-11-01804-f005:**
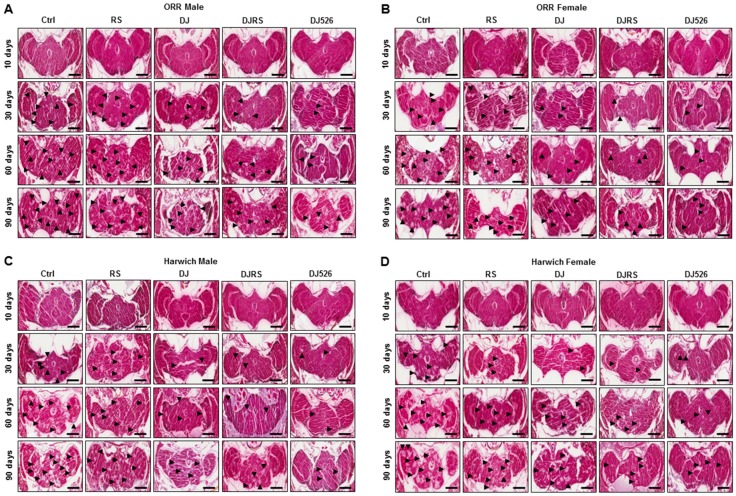
The resveratrol rice DJ526 inhibited age-related neurodegeneration in *D. melanogaster*. (**A**) ORR males, (**B**) ORR females, (**C**) Harwich males and (**D**) Harwich females. Ctrl represents standard cornmeal medium; RS represents cornmeal medium supplemented with resveratrol at 31.54 µg/L, the equivalent amount of resveratrol found in the DJ526 media; DJ represents the medium in which 50% of cornmeal was replaced with Dongjin rice; DJRS represents the medium in which 50% of cornmeal was replaced with Dongjin rice and supplemented with resveratrol at 31.54 µg/L, and DJ526 represents the medium in which 50% of the cornmeal was replaced with the resveratrol rice DJ526 (Appendix A). A histological analysis was performed by H&E staining to examine the neurodegeneration of the *Drosophila* brains at the 10th, 30th, 60th and 90th days post-eclosion. *n* = 100; scale bars: 200 μm. The vacuolar lesions are indicated as black triangles.

**Figure 6 nutrients-11-01804-f006:**
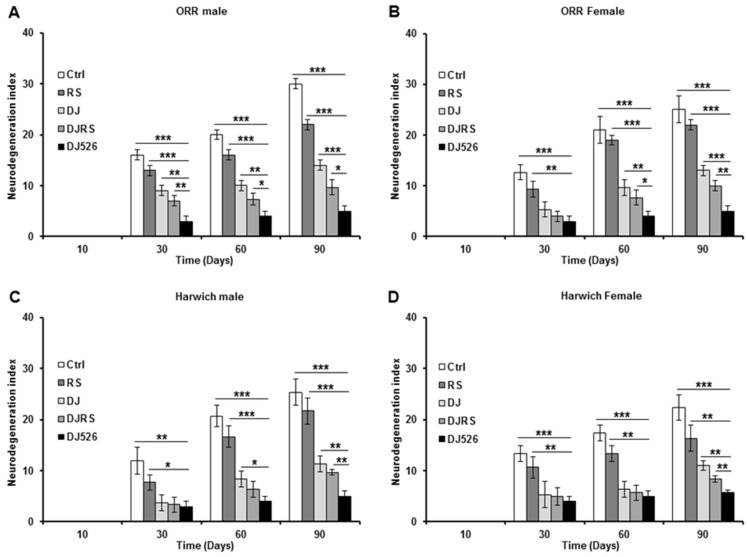
The resveratrol rice DJ526 suppressed age-related neurodegeneration in *D. melanogaster*. (**A**) ORR males, (**B**) ORR females, (**C**) Harwich males and (**D**) Harwich females. Ctrl represents standard cornmeal medium; RS represents cornmeal medium supplemented with resveratrol at 31.54 µg/L, the equivalent amount of resveratrol found in the DJ526 media; DJ represents the medium in which 50% of cornmeal was replaced with Dongjin rice; DJRS represents the medium in which 50% of cornmeal was replaced with Dongjin rice and supplemented with resveratrol at 31.54 µg/L, and DJ526 represents the medium in which 50% of the cornmeal was replaced with the resveratrol rice DJ526 (Appendix A). The quantification of the neurodegeneration and vacuolar lesions based on the histological analysis of the *Drosophila* brains were observed at the 10th, 30th, 60th and 90th days post-eclosion. Statistical significance was analysed with an unpaired Student’s *t*-test and indicated as * *p* < 0.05, ** *p* < 0.01, and *** *p* < 0.001 from three independent experiments *(n* = 30).

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
