# Peer review of "Effect of the Resveratrol Rice DJ526 on Longevity"

_nutrients, 2019, doi:10.3390/nu11081804_

Round 1
Reviewer 1 Report
Authors present results indicating lifespan extension and protection against some forms of age-related degeneration by feeding Drosophila a rice strain that produces resveratrol. Results appear sound, however manuscript should be adjusted to make clear the contribution to the field and to provide more clarity to the interpretation of the results. One additional control experiment should also be added.
Major concerns -
Authors need to do more in the text to explain the difference between these experiments and the recently published, very similar paper that used DJ526 rice callas. It is not obvious what is different, or why the new experiments add to our understanding.
Authors do not report controls for feeding rate or total caloric intake between the diets. This is important because if caloric intake is different, this could affect the results in a way that is independent of the DJ526 rice. Some measurement of intake of flies on the different diets should be included.
Minor concerns -
50 flies per vial is a little bit on the crowded side for longevity measurements. 20 is more standard. Not necessarily a major problem, but something to consider.
Citation is needed for the claim that 18C is optimal temperature for flies. Not sure what is meant by this statement. They will certainly live longer than at higher temperatures, but will also reproduce slower, and have other tradeoffs.
Citation is also needed for the claim that flies with lower weight are maintaining "healthier weight". I am not familiar with work that correlates weight with health indices in flies, and the cites here refer to mammals, where the connection is more clear. This is significant, because reduced weight could indicate reduced caloric intake, and this could suggest indirect effects from caloric restriction are playing a role in the effects of DJ526.
Author Response
Point-by-point response to Reviewer #1
We appreciate the time and efforts by the editor and reviewer of Nutrient in reviewing our manuscript. We corrected and answered all of points addressed by the reviews, and believed that the revised version can meet the publication requirements of Nutrient. We are deeply thankful for your helpful comments and suggestions. These comments have helped us a lot in clarifying and sharpening our manuscript. Detailed corrections are listed as below.
Comments and Suggestions for Authors:
Authors present results indicating lifespan extension and protection against some forms of age-related degeneration by feeding Drosophila a rice strain that produces resveratrol. Results appear sound, however manuscript should be adjusted to make clear the contribution to the field and to provide more clarity to the interpretation of the results. One additional control experiment should also be added.
Response to Comment:
We corrected our manuscript based on your suggestion including author’s contribution, interpretation of the results and added another control data in the figures (when we performed the experiments, the controls were included. However, we omitted these controls. I agree your opinion to include these controls, and very appreciate your thoughtful comment). We appreciate the time and efforts by reviewer in reviewing this manuscript. Thank you.
Major concerns
Comment #1:
Authors need to do more in the text to explain the difference between these experiments and the recently published, very similar paper that used DJ526 rice callas. It is not obvious what is different, or why the new experiments add to our understanding.
Response to Comment:
We described our previous publication in the manuscript (page 2) based on your suggestion. We really appreciate your proper comment. Thank you.
Comment #2:
Authors do not report controls for feeding rate or total caloric intake between the diets. This is important because if caloric intake is different, this could affect the results in a way that is independent of the DJ526 rice. Some measurement of intake of flies on the different diets should be included.
Response to Comment:
The feeding amount or total caloric intake seemed not much different among different groups. In mammals, food ingestion can be directly quantified by weighing the food before and after diet. However, flies consume foods that are too little to weigh accurately, and feed by extension of their proboscis into the food medium, prohibiting direct observation of the volume of food ingested. Since we were not sure about the reliability of the feed intake, we designed various controls, including media control, resveratrol control, DJ control as well as DJRS. Following your valuable suggestion, we added the original DJRS data which we did not include in the original manuscript. Thank you.
Minor concerns
Comment #1:
50 flies per vial is a little bit on the crowded side for longevity measurements. 20 is more standard. Not necessarily a major problem, but something to consider.
Response to Comment:
You are absolutely right but should not be concerned. In this study, we used a big plastic vial which can easily maintain 50 flies and thus it was not a crowded condition for longevity measurements. We appreciate your proper comment. Thank you.
Comment #2:
Citation is needed for the claim that 18C is optimal temperature for flies. Not sure what is meant by this statement. They will certainly live longer than at higher temperatures, but will also reproduce slower, and have other tradeoffs.
Response to Comment:
We corrected our manuscript based on your suggestion (page-3). We appreciate your valuable suggestions. Thank you.
Comment #3:
Citation is also needed for the claim that flies with lower weight are maintaining "healthier weight". I am not familiar with work that correlates weight with health indices in flies, and the cites here refer to mammals, where the connection is more clear. This is significant, because reduced weight could indicate reduced caloric intake, and this could suggest indirect effects from caloric restriction are playing a role in the effects of DJ526.
Response to Comment:
Previously, we showed that the effect of resveratrol rice DJ526 on body weight in mice with age-progression [22] and DJ526 callus in D. melanogaster [25]. Under the same caloric intake, this could be explained as the beneficial effect of DJ526 on body weight of D. melanogaster during age-progression. Thank you.
Again, we appreciate your valuable suggestions. Please let me know if you require any further information.
Best regards,
Seong-Tshool Hong, PhD
Reviewer 2 Report
The authors sealed with the effect of a vegetal flavonoid and anti-oxydant , resveratrol, in longevity in D. melanogaster. The study is in adequation with former studies of the group which created a novel specie of rice (DJ526), able to express resveratrol, and tested as food to investigate the role of this flavonoid in animals. The study proposed is related to a novel food-mediated way to prevent or delay the deleterious effect of ageing.
This article is well-written and the objectives are clear since nice transition and explanations are proposed as transitions between each experimental part.
The paper is acceptable is this format, however I can suggest some slight modifications to make some figures clearer for the readers. Figures 5 and 6 have to be bigger, and concerning Figure 5, some pictures are difficult to evaluate since the brightness is sometimes heterogenous.
Author Response
Point-by-point response to Reviewer #2
We appreciate your time and efforts in reviewing our manuscript. Your comments have helped us in improving our manuscript. Following your suggestion, we have modified the manuscript accordingly as shown below.
Comments and Suggestions for Authors:
The authors sealed with the effect of a vegetal flavonoid and anti-oxydant, resveratrol, in longevity in D. melanogaster. The study is in adequation with former studies of the group which created a novel specie of rice (DJ526), able to express resveratrol, and tested as food to investigate the role of this flavonoid in animals. The study proposed is related to a novel food-mediated way to prevent or delay the deleterious effect of ageing.
This article is well-written and the objectives are clear since nice transition and explanations are proposed as transitions between each experimental part.
The paper is acceptable is this format, however I can suggest some slight modifications to make some figures clearer for the readers. Figures 5 and 6 have to be bigger, and concerning Figure 5, some pictures are difficult to evaluate since the brightness is sometimes heterogenous.
Response to Comment:
We enlarged the size of Figure 5 and 6 following your suggestion (page-9, 10). We appreciate your time and efforts in reviewing this manuscript. Thank you.
Again, we appreciate your valuable suggestions. Please let me know if you require any further information.
Best regards,
Seong-Tshool Hong, PhD
Round 2
Reviewer 1 Report
Authors have included an additional control that addresses my concerns about feeding rate. Since a feeding rate change would also affect this group if it existed, this satisfies me. However, the authors should take note for the future that there are several good ways to measure feeding rate in flies that are not very difficult and are often required for publication. These include measuring frequency of proboscis extension or feeding a dye and measuring dye in the gut.